# Speak, Edit, Repeat: High-Fidelity Voice Editing and Zero-Shot TTS with Cross-Attentive Mamba

## Abstract

We introduce **MAVE** (**M**amba with Cross-**A**ttention for **V**oice **E**diting and Synthesis), a novel autoregressive architecture for text-conditioned voice editing and high-fidelity text-to-speech (TTS) synthesis, built on a cross-attentive Mamba backbone. MAVE achieves state-of-the-art performance in speech editing and very competitive results in zero-shot TTS, while not being explicitly trained on the latter task, outperforming leading autoregressive and diffusion models on diverse, real-world audio. By integrating Mamba for efficient audio sequence modeling with cross-attention for precise text-acoustic alignment, MAVE enables context-aware voice editing with exceptional naturalness and speaker consistency. In pairwise human evaluations on a random 40-sample subset of the RealEdit benchmark (400 judgments), 57.2% of listeners rated MAVE-edited speech as perceptually equal to the original, while 24.8% prefered the original and 18.0% MAVE- demonstrating that in the majority of cases edits are indistinguishable from the source. MAVE compares favorably with VoiceCraft and FluentSpeech both on pairwise comparisons and standalone mean opinion score (MOS) evaluations. For zero-shot TTS, MAVE exceeds VoiceCraft in both speaker similarity and naturalness, without requiring multiple inference runs or post-processing. Remarkably, these quality gains come with a significantly lower memory cost and approximately the same latency: MAVE requires $\sim 6\times$ less memory than VoiceCraft during inference on utterances from the RealEdit database (mean duration: 6.21s, A100, FP16, batch size 1). Our results demonstrate that MAVE establishes a new standard for flexible, high-fidelity voice editing and synthesis through the synergistic integration of structured state-space modeling and cross-modal attention.

## 1 Introduction

Recent advances in neural speech synthesis have enabled compelling text-to-speech (TTS) and voice editing capabilities, yet *precise, context-aware voice editing* remains a formidable challenge. Current approaches fall into two paradigms: **autoregressive models** (AR) such as Transformer-based methods Peng et al. (2024) and **non-autoregressive frameworks** (NAR) that are based on flow-matching or diffusion probabilistic models Le et al. (2023); Guo et al. (2024); Du et al. (2024)). While AR models excel in fidelity, they suffer from *quadratic complexity*. Conversely, flow-matching methods Le et al. (2023); Guo et al. (2024) prioritize speed, but struggle with *temporal coherence* and *fine-grained prosodic control*, particularly in noisy, real-world audio.

To address these limitations, we propose **MAVE** (**M**amba with Cross-**A**ttention for **V**oice **E**diting and Synthesis), a novel *autoregressive* architecture for text-conditioned voice editing and zero-shot TTS that synergizes Mamba state-space models with cross-attention mechanisms. Unlike Transformer-based decoders Peng et al. (2024); Wang et al. (2023a), MAVE replaces self-attention with structured state-space sequences (SSMs), enabling *linear-complexity modeling* of dependencies between acoustic tokens. Crucially, our cross-attention module dynamically aligns *augmented text inputs* with acoustic tokens, allowing the model to "edit" speech by attending to the textual information.

To the best of our knowledge, **MAVE** – built on a cross-attentive Mamba backbone – is the first successful application of a structured state-space model to text-conditional speech generation,

namely speech editing and zero-shot TTS. On the challenging RealEdit benchmark, MAVE achieves **human-parity naturalness** in speech editing and surpasses state-of-the-art models like VoiceCraft and FluentSpeech in both speaker similarity and naturalness (MOS) without requiring any post-processing. Moreover, MAVE offers significant efficiency gains, reducing memory usage by $6\times$ compared to Transformer-based VoiceCraft, while enabling single-pass generation without silent tokens handling or multiple runs. Our results demonstrate that Mamba-based models enhanced with cross-attention can outperform both autoregressive and diffusion-based approaches in fidelity, efficiency, and robustness, establishing a new direction for scalable and high-quality speech generation.

## 2 RELATED WORKS

**Neural Codec Language Models for Speech Synthesis** The evolution of high-fidelity speech generation has been significantly advanced by Neural Codec Language Models (NCLMs), which discretize speech into symbolic units through Residual Vector Quantization (RVQ) frameworks as shown by Zeghidour et al. (2021); Defossez et al. (2022) and model temporal dependencies via autoregressive sequence learning. Originating in textless NLP research Hsu et al. (2021); Lakhotia et al. (2021); Kharitonov et al. (2021); Nguyen et al. (2022), this paradigm achieved breakthrough performance in speech continuity with AudioLM Borsos et al. (2022a). A pivotal advancement emerged through the application of NCLMs to zero-shot text-to-speech synthesis, where models like VALL-E Wang et al. (2023a) and Spear-TTS Kharitonov et al. (2023) reframed synthesis as transcript-conditioned speech continuation conditioned on brief speaker references. These systems substantially outperformed conventional non-NCLM approaches, prompting extensions for cross-lingual TTS Zhang et al. (2023), style-controlled synthesis Guo et al. (2022); Yang et al. (2023); Liu et al. (2023); Ji et al. (2023); Lyth & King (2024), and phoneme alignment refinement Song et al. (2025).

**The Evolution of Speech Editing** Speech editing, defined as the precise modification of targeted speech segments while preserving unaltered regions, has evolved through three distinct generations of methodology. Early approaches Jin et al. (2017) relied on concatenating TTS-generated segments with original speech, inevitably introducing prosody mismatches and boundary artifacts due to the absence of contextual conditioning Morrison et al. (2021). Subsequent research introduced context-aware mechanisms through bidirectional fusion architectures Tan et al. (2021) and masked reconstruction objectives Wang et al. (2022); Bai et al. (2022); Borsos et al. (2022b), with Transformer-based systems demonstrating improved contextualization. Most recently, diffusion-based methods like FluentSpeech Jiang et al. (2023) achieved state-of-the-art performance on standard benchmarks through denoising frameworks. However, these approaches collectively suffer from three interrelated limitations. First, Transformer-based systems incur quadratic computational complexity relative to sequence length, restricting practical context windows. Second, evaluation protocols remain constrained to short editing spans—UniCATS Du et al. (2024), for instance, limits assessments to segments under two seconds, failing to address real-world editing scenarios involving multi-word phrases. Third and most critically, none incorporate mechanisms for leveraging linguistically augmented inputs (e.g., prosody-annotated text) to guide context-aware modifications. MAVE overcomes these constraints through its linear-complexity Mamba architecture and differentiable cross-attention fusion mechanism, enabling robust editing of spans up to 16 words while preserving natural prosody through explicit text augmentation.

**Unified Frameworks for Voice Editing and Synthesis** Recent efforts have sought to develop unified models capable of both zero-shot TTS and speech editing, recognizing their shared dependency on context-aware acoustic modeling. These frameworks broadly bifurcate into modular architectures Yin et al. (2022); Jiang et al. (2023) that employ separate components for distinct tasks, and end-to-end systems including SpeechX Wang et al. (2023b)—which adapts VALLE through prompt tuning—and flow-matching approaches like VoiceBox Le et al. (2023) and UniCATS Du et al. (2024). While representing important conceptual advances, these unified systems exhibit significant practical limitations. SpeechX lacks human evaluation metrics for editing performance, undermining claims of perceptual quality. VoiceBox, despite its broad task coverage, omits formal speech editing evaluation in its methodology, relying solely on demo examples. UniCATS restricts editing capability to brief segments ($< 2$ seconds) and employs rule-based prosody markers rather than differentiable conditioning. Crucially, all existing unified frameworks are constrained by a fundamental dichotomy: autoregressive systems (e.g., SpeechX) maintain high fidelity but sacrifice contextual awareness through causal masking and face quadratic time complexity, while

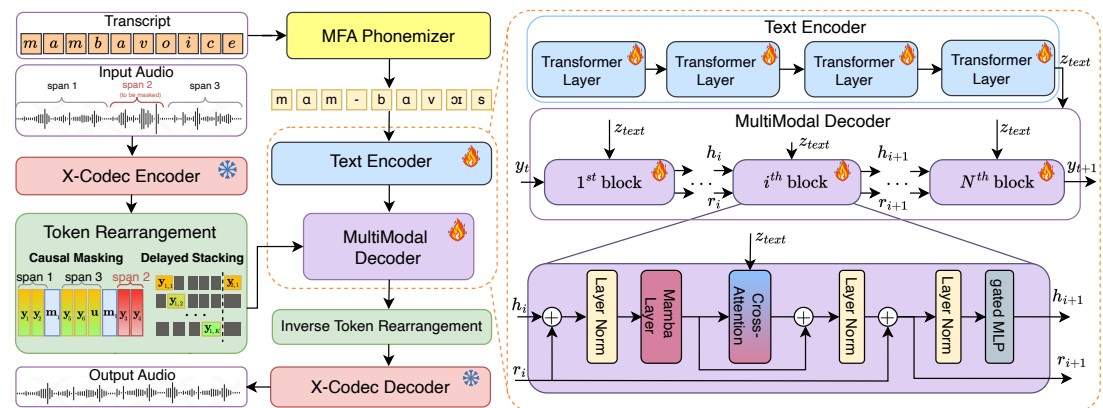

Figure 1: Overview of the proposed MAVE architecture. The model accepts phonemized text and audio tokens as input. A causal masking and rearrangement strategy is applied to the audio tokens to enable bidirectional context for editing. The core of the model is a Mamba block for efficient sequence modeling, augmented with cross-attention layers to condition the audio generation on the text embeddings produced by a Transformer encoder.

non-autoregressive approaches (e.g., VoiceBox) enable bidirectional context modeling at the cost of temporal coherence and prosodic precision. This trade-off between contextual awareness and generation fidelity has remained unresolved in prior art.

**MAVE in a Nutshell** In summary, MAVE establishes a novel paradigm in text-conditioned speech generation by bridging the efficiency of State Space Models with the flexibility of cross-modal attention. Unlike prior SSM-based approaches such as CM-Mamba Huang et al. (2024), which require strict alignment between text and audio sequences, our architecture introduces a length-agnostic cross-attention mechanism that enables rich, contextualized phoneme conditioning without artificial upsampling or architectural constraints. By replacing the quadratic self-attention of Transformer-based editors like Voicecraft with linear-time selective state updates, MAVE achieves scalable long-context modeling while preserving prosodic coherence and speaker identity through recurrence. Our framework further unifies three critical capabilities: (1) efficient bidirectional context access via causal token rearrangement, (2) precise linguistic control through differentiable cross-attention on phonemized input, and (3) reference-guided voice consistency without explicit speaker embeddings. As demonstrated in Section 4, MAVE is the first model to simultaneously outperform both autoregressive (e.g., VoiceCraft) and non-autoregressive (e.g., FluentSpeech) baselines in human evaluations across speech editing and zero-shot TTS, setting a new standard for unified, high-fidelity, and scalable speech generation. To our knowledge, this is the first successful integration of cross-attention into a Mamba-based audio decoder for unrestricted, long-form speech editing and synthesis.

## 3 PROPOSED METHOD

We present **MAVE**, a novel architecture for text-conditioned speech editing and zero-shot text-to-speech (TTS). The core challenge lies in autoregressively generating long sequences of discrete audio tokens conditioned on a textual transcript, where the quadratic complexity of self-attention in Transformers becomes prohibitive. Audio signals are typically encoded at 50 Hz using residual vector quantization (RVQ), yielding 4 to 8 discrete tokens per frame, which amounts to 200–400 tokens per second of speech Defossez et al. (2022); Ye et al. (2025). In contrast, the corresponding phoneme sequence contains approximately 12 units per second Sigurd et al. (2004); Balota et al. (2007); Santi et al. (2024), resulting in a significant sequence length mismatch between modalities. This disparity makes direct application of Transformer-based models inefficient for high-fidelity, long-form audio generation.

While State Space Models (SSMs) such as Mamba Gu & Dao (2023) offer linear-time sequence modeling, they face two key challenges in multimodal speech generation: (1) limited support for cross-modal conditioning (such as encoder-decoder architecture of transformers), and (2) potential

loss of fine-grained acoustic details over long sequences. Prior work such as CM-Mamba Huang et al. (2024) enables cross-attention but requires equal-length modalities, which is infeasible for text-conditioned speech processing. Unlike CM-Mamba, which requires equal-length modalities and thus forces phoneme padding or audio subsampling, our architecture supports arbitrary-length text-to-audio conditioning via explicit Cross-Attention module, attended access to text embeddings, decoupled from the autoregressive audio state evolution.

To address these issues, we propose MAVE, a hybrid decoder that combines the efficiency of Mamba for audio token modeling with a flexible cross-attention mechanism for text conditioning—without requiring aligned sequence lengths. Our model supports context-aware speech infilling and zero-shot TTS by leveraging both surrounding audio context and linguistic input.

### 3.1 Input Representation and Token Rearrangement

Given an input audio waveform $X \in \mathbb{R}^{T \times f_s}$, where $T$ is duration in seconds and $f_s$ is the sampling rate (e.g., 16 kHz), we encode it using X-Codec, a residual vector quantization (RVQ)-based neural codec Ye et al. (2025). This yields a sequence of $K = 8$ parallel streams of discrete tokens at a frame rate of $f_x = 50$ Hz. Let $L = T \cdot f_x$ denote the number of frames. The encoded representation is:

$$Y = [\mathbf{y}_1, \mathbf{y}_2, \ldots, \mathbf{y}_L], \quad \text{where } \mathbf{y}_l = [y_{l1}, y_{l2}, \ldots, y_{lK}], \quad y_{lk} \in \{1, 2, \ldots, S\},$$

and $S = 1024$ is the codebook size per RVQ level.

For speech editing, we consider a non-autoregressive infilling task where one or more spans of audio tokens are masked and must be reconstructed. To enable bidirectional context access during autoregressive generation, we adopt the **causal masking** strategy from CM3 Aghajanyan et al. (2022). Masked spans are replaced with special mask tokens and moved to the end of the sequence, allowing the model to condition on both past and future context while preserving autoregressive training.

During training, we sample $m \sim \text{Poisson}(\lambda = 1)$ masked spans, capped at a maximum of $N = 3$. Each span length is uniformly sampled from $[1, 600]$ frames, and positions are selected to avoid overlap. If the $j$-th span $s_j$ is masked, we:

1. Replace the span with a unique mask token $M_j$,
2. Append a mask token $M_j$, followed by the original token sequence $s_j$.

For example, given $Y = [\mathbf{s}_1, \mathbf{s}_2, \mathbf{s}_3, \mathbf{s}_4, \mathbf{s}_5]$ and masking spans $\mathbf{s}_2, \mathbf{s}_4$, the rearranged sequence becomes:

$$Y_{\text{rearranged}} = [\mathbf{s}_1, M_1, \mathbf{s}_3, M_2, \mathbf{s}_5, M_1, \mathbf{s}_2, M_2, \mathbf{s}_4].$$

The model autoregressively generates tokens following the last $M_1$, reconstructing the original masked content. After that, to accelerate decoding, we apply the **codebook delay pattern** Kharitonov et al. (2021); Copet et al. (2023). This is mainly to ensure that the generation of a speific code of level $k$ at some time step $t$ is conditioned on all the higher levels $[1, ..., k-1]$ at the same time step $t$.

### 3.2 Modeling and Text Conditioning

Having defined the input representation and preprocessing pipeline, we now describe how the model leverages Mamba and cross-attention to generate audio tokens autoregressively conditioned on text and reference context. The task is formulated as *text-conditioned autoregressive audio generation*, decomposed into two components: (1) modeling intra-audio temporal dependencies, and (2) conditioning on linguistic input.

#### 3.2.1 Modeling Intra-Audio Token Dependencies

We employ the Mamba architecture Gu & Dao (2023), a selective State Space Model (SSM), to model long-range dependencies in the audio token sequence. Unlike Transformers, Mamba scales linearly with sequence length and maintains a compressed latent state that effectively captures speaker identity, prosody, and acoustic continuity—critical for coherent speech editing.

The SSM operates via a discretized recurrence with input-dependent parameters. At step $t$, given input $x_t \in \mathbb{R}^d$, the hidden state $s_t \in \mathbb{R}^d$ evolves as:

$$s_t = \overline{A}_t s_{t-1} + \overline{B}_t x_t, \quad \text{(1a)} \qquad\qquad g_t = \overline{C}_t s_t, \quad \text{(1b)}$$

where $\overline{A}_t, \overline{B}_t, \overline{C}_t$ are discretized SSM parameters computed from $x_t$ via a shared projection. The output $g_t$ is then passed through a gating mechanism (e.g., SiLU) and combined with a residual branch. This selective mechanism allows Mamba to dynamically attend to relevant past information, making it well-suited for preserving voice and prosody across long contexts.

We stack multiple Mamba blocks with intermediate normalization and residual connections, forming the backbone of our audio decoder. Mamba has proven to be very effective in terms of modeling dependencies between audio tokens in previous research Erol et al. (2024) Shams et al. (2024), but its effectivness to generate text-conditioned conherent audio are still not well-studied. Many text-conditioned audio generation approaches, such as text-to-speech (TTS) and speech editing, have adopted transformer-based decoder architectures that process concatenated text and audio tokens within a unified sequence Peng et al. (2024) Wang et al. (2023a). While this design leverages the strong sequence modeling capabilities of transformers, it poses significant challenges when applied to alternative architectures such as Mamba.

Unlike transformers, which benefit from global self-attention, Mamba relies on selective state-space mechanisms that exhibit difficulty in retaining fine-grained information over long sequences. As a result, if we concatenate text and audio tokens and attempt to process them with the same decoder model, distant text tokens—critical for maintaining linguistic fidelity—tend to be encoded with degraded or "fuzzy" memory Waleffe et al. (2024), leading to a loss of semantic precision. This limitation is particularly detrimental in speech generation tasks, where high-fidelity retention of textual information is essential to ensure accurate alignment and high-quality output.

On the other hand, this limitation is less critical when modeling dependencies among audio tokens, as exact low-level detail preservation is not essential for high-quality speech synthesis. Instead, the model benefits from retaining high-level acoustic features which are sufficient to generate coherent and natural-sounding audio. The compressed, selective memory inherent in SSMs like Mamba is well-suited to this purpose, effectively filtering out perceptually irrelevant noise while preserving meaningful temporal patterns. In our ablation studies, we further demonstrate that conditioning on text via cross-attention—rather than token concatenation—is crucial for maintaining linguistic fidelity in Mamba-based architectures. This design ensures that textual information is explicitly attended to throughout generation, mitigating the risk of forgetting distant linguistic context.

### 3.2.2 TEXT CONDITIONING VIA CROSS-ATTENTION

To incorporate linguistic information, we first convert the input transcript into a sequence of phonemes using the Montreal Forced Aligner (MFA) toolkit McAuliffe et al. (2017). This ensures explicit modeling of pronunciation, especially for homographs. The phoneme sequence is then processed by a 4-layer Transformer encoder to produce contextualized embeddings $\mathbf{z}_{\text{text}} = [\mathbf{z}_1, \ldots, \mathbf{z}_M] \in \mathbb{R}^{M \times d}$, where $M$ is the number of phonemes. As depicted in Fig. 1, a cross-attention module is inserted after each Mamba block. The query is derived from the Mamba block's output, while keys and values come from $\mathbf{z}_{\text{text}}$. This allows the audio decoder to attend to relevant phonetic content at each generation step, ensuring precise linguistic alignment.

### 3.2.3 SPEAKER CONDITIONING VIA REFERENCE CONTEXT

MAVE does not rely on explicit speaker embeddings for voice identity preservation. Instead, it leverages *in-context learning* for TTS task by prepending a short reference utterance encoded into discrete audio tokens using `X-Codec` Ye et al. (2025) to the generation sequence. This reference context provides rich acoustic, prosodic, and timbral cues that guide the autoregressive decoder to synthesize speech in the target speaker's voice, without requiring speaker labels or fine-tuning.

During both training and inference, the reference tokens are placed at the beginning of the input sequence, followed by the masked audio context. The Mamba decoder attends to this context through its long-range recurrence, effectively conditioning the generated speech on the speaker's voice characteristics. This strategy enables true zero-shot TTS: given only a few seconds of reference audio and a target transcript, the model generates high-fidelity, speaker-consistent speech.

In speech editing tasks, the reference can be omitted because the surrounding unmasked audio already provides sufficient speaker context. However, for zero-shot TTS or cross-speaker editing, the reference utterance is essential for voice coherence.

**Training and Inference.** Given the input context (including text, reference audio, and unmasked audio tokens), the model parametrized by $\theta$ autoregressively predicts the conditional distribution over the RVQ codebooks.

We define the training loss as the negative log-likelihood:

$$\mathcal{L}(\theta) = -\log P_\theta(\mathbf{Y} \mid \mathcal{C}) = -\sum_{k=1}^{K} \mathcal{L}_k(\theta), \tag{2}$$

where $\mathcal{C}$ denotes the conditioning transcript, and $\mathcal{L}_k(\theta) = \sum_{l=1}^{L} \log P_\theta(y_{lk} \mid \mathbf{y}_{<l}, \mathbf{y}_{l,<k}, \mathcal{C})$ is the cross-entropy loss for the $k$-th codebook.

The early RVQ codebooks in `X-Codec` Ye et al. (2025) encode semantic and linguistic content, while the later codebooks primarily model fine-grained acoustic details and high-frequency textures. To prioritize the accurate reconstruction of perceptually important features, we employ a weighted loss strategy:

$$\mathcal{L}(\theta) = -\sum_{k=1}^{K} \alpha_k \mathcal{L}_k(\theta), \tag{3}$$

where $\{\alpha_k\}_{k=1}^{K}$ are tunable hyperparameters with $\alpha_1 \geq \alpha_2 \geq \cdots \geq \alpha_K$, typically set inversely proportional to codebook index. Following Liu et al. (2025) we assign a weight of 0.25 to the first three levels and 0.05 to the last five. Additionally, following Aghajanyan et al. (2022), we compute the prediction loss over all valid audio tokens in the sequence, excluding special tokens such as mask tokens $M_j$ regardless of whether they belong to masked or unmasked regions. This encourages the model to refine its internal representations throughout the sequence, improving contextual coherence and reconstruction fidelity.

In summary, MAVE is a hybrid autoregressive decoder that integrates Mamba blocks for efficient long-sequence audio modeling with cross-attention for flexible text conditioning. By combining token rearrangement, delayed RVQ decoding, and explicit speaker conditioning, it supports high-fidelity speech editing and zero-shot TTS. The network efficiently handles a large disparity between text and audio sequence lengths, offering a scalable alternative to transformer-based methods.

## 4 EXPERIMENTS

**Dataset.** Gigaspeech training set Chen et al. (2021) is used as the training data, which contains 9k hours of audiobooks, podcasts, and YouTube videos at 16kHz audio sampling rate. For speech editing evaluation, we use the real-world benchmark called RealEdit from Peng et al. (2024). For TTS evaluation, we use a subset of libritts Zen et al. (2019b)

**Training Details.** To train MAVE, we used the ScaledAdam optimizer and Eden Scheduler proposed in Yao et al. (2024) with a base learning rate of 0.01, batch size of 400k frames (i.e. 133.2 minutes), and total training step of 50k with gradient accumulation. The training of the 830M MAVE model took about 4 days on 4 NVIDIA A100 GPUs. For inference, we use Nucleus sampling Holtzman et al. (2020) with p = 0.8 and a temperature of 1 for all experiments.

**Evaluation Setup** VoiceCraft has been observed to generate prolonged silences during synthesis Peng et al. (2024). To mitigate this issue, the authors proposed reducing the probability of silence tokens and performing multiple generations per sample, selecting the shortest output as the final result. In all of our experiments involving VoiceCraft, we adopted this strategy, generating five samples per input and selecting the shortest one. In contrast we haven't noticed this problem in our model, so we produce a single generation with no further processing.

### 4.1 RESULTS ON SPEECH EDITING

Table 1 shows the results of both VoiceCraft Peng et al. (2024) and MAVE on RealEdit dataset Peng et al. (2024), which reflects real-life speech editing cases that are both challenging and diverse. The original dataset contains 100 utterances from LibriTTS (dev-clean and dev-other) Zen et al. (2019a),

Table 1: Results on RealEdit benchmark for the speech editing task.

| Model | WER (L) ↓ | WER (M) ↓ | MOS Naturalness ↑ | MOS Intelligibility ↑ |
|---|---|---|---|---|
| VoiceCraft | 8.4 | 6.9 | 3.77 ± 0.08 | 4.20 ± 0.07 |
| MAVE (ours) | **7.5** | **5.9** | **3.90 ± 0.08** | **4.25 ± 0.07** |
| Ground Truth | 6.8 | 5.2 | 4.00 ± 0.08 | 4.31 ± 0.06 |

100 utterances from YouTube (from Gigaspeech testset) Chen et al. (2021), and 110 utterances from the Spotify Podcast dataset Clifton et al. (2020). However, the Spotify podcasts dataset is no longer available on the official website, so we have used only the 200 audios that are still publicly available.

We computed the Word Error Rate (WER) between the generated audio and the target transcript using Whisper-large and Whisper-medium.en Radford et al. (2022a). In addition, we conducted a human evaluation to assess naturalness and intelligibility, based on audio samples from 80 randomly selected test cases for each model. Specifically, 20 people of proven English fluency participated in the study, they were divided into 2 groups of 10 people, and each group was asked to evaluate 120 audios (40 from each method) on the naturalness and intelligibility on a 5-point Likert scale (poor=1, excellent=5). For further details we refer to A.3.1.

MAVE demonstrates superior performance compared to VoiceCraft, achieving smaller WER and higher mean opinion scores (MOS) in both naturalness (3.90 vs. 3.77) and intelligibility (4.25 vs. 4.20). Notably, the performance gap between our model and the ground truth recordings is minimal, with differences of less than 0.1 in naturalness MOS and less than 0.07 in intelligibility MOS. This is an indication that MAVE-edits feature strong perceptual correlation to human speech. It is worth noting that even for the original audio, the average naturalness ratings did not exceed 4.0 out of 5.0. This is mostly attributed to the fact that the audio data were collected in the wild and their quality is inferior to studio recordings, with the presence of background noise and other sources of recording artifacts. This also highlights the challenging conditions under which the evaluation was conducted.

To further assess the perceptual quality of our generated speech relative to ground truth, we conducted an additional study in which we asked a group of 10 users of proven English fluency to do a side-by-side comparison on the naturalness of the original unedited audio and our edited audio on 40 samples from the RealEdit dataset. The results are very promising and demonstrate that 57.2% of the times, users reported that both audios sound equally natural, while in 24.8% of cases, the ground truth was judged as more natural, and interestingly 18% of the times, the edited audio was preferred over the original. These findings indicate that in the majority of cases, our generated audios are indistinguishable from the original ones. For further details about the setup, we refer to A.3.2

To enable a meaningful comparison with diffusion-based speech generation models, we selected FluentSpeech Jiang et al. (2023) as the most relevant open-source baseline. However, due to its significantly smaller model size, a direct comparison with our approach would not be fair. To address this, we leveraged the larger variant of FluentSpeech retrained and evaluated by the authors of Voice-Craft. The authors have not made this model publicly available but have provided 14 synthesized utterances, publishing for each sample: (1) VoiceCraft's generation, (2) the enhanced FluentSpeech model's generation, and (3) the original audio with original and target transcript. We generated corresponding outputs using MAVE for the same 14 samples and conducted a perceptual listening study to compare the audio quality between the three models. The evaluation consisted of a pairwise, side-by-side comparison, where each participant (20 in total) was presented with two audio samples from different models and asked to judge which exhibited better naturalness and intelligibility. With 14 samples per system and three possible pairings between the models (VoiceCraft vs. FluentSpeech, VoiceCraft vs. Ours, and FluentSpeech vs. Ours), each participant evaluated a total of 42 audio pairs. Further details on the experimental setup, including participant instructions, interface design, and evaluation protocol, are provided in Appendix A.3.2.

The results in Figure 2 clearly indicate that MAVE accomplishes superior perceptual quality compared to both FluentSpeech and VoiceCraft. While VoiceCraft already holds an advantage over FluentSpeech as it was preferred in 47.1% vs. 13.2% for naturalness and 50.7% vs. 6.4% for intelligibility, our model surpasses FluentSpeech by an even larger margin, being favored in 62.9% of cases for naturalness and 59.3% for intelligibility. More importantly, our model also outperforms

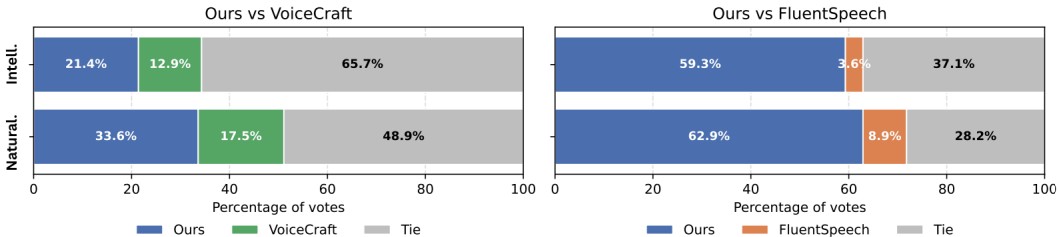

Figure 2: Side-by-side comparison between MAVE (ours), VoiceCraft and FluentSpeech

Table 2: Performance analysis of the proposed MAVE on the zero-shot TTS task.

| Model | WER (L) ↓ | WER (M) ↓ | SIM ↑ | MCD ↓ | MOS Naturalness ↑ | MOS Intelligibility ↑ |
|---|---|---|---|---|---|---|
| VoiceCraft | 9.3 | 7.5 | 0.55 | 4.75 | 3.22 ± 0.07 | 4.01 ± 0.07 |
| MAVE (ours) | **7.4** | **6.6** | **0.57** | **4.73** | **3.48 ± 0.08** | **4.20 ± 0.06** |
| Ground Truth | 6.2 | 5.4 | 0.66 | - | 3.90 ± 0.08 | 4.40 ± 0.06 |

VoiceCraft: it is preferred over VoiceCraft in 33.6% of comparisons for naturalness and 21.4% for intelligibility, while VoiceCraft is preferred over ours in only 17.5% and 12.9%, respectively.

## 4.2 RESULTS ON ZERO-SHOT TTS

For zero-shot text-to-speech (TTS) evaluation, we used the LibriTTS dataset Zen et al. (2019a) and we randomly selected 500 utterances. For each utterance, we extracted a different uterrance with the same speaker ID. The prompt was trimmed to be as close as possible to 3 seconds in duration, with cuts made only at word boundaries. We then filtered the initial 500 samples to retain only those containing more than 8 words in the target text, ensuring sufficient linguistic complexity for meaningful evaluation. This filtering resulted in a final evaluation set of 372 samples.

Table 2 shows the results for the zero-shot TTS task. For the evaluation we employed both objective and subjective metrics. The objective evaluation included WER, computed using Whisper Large and Whisper medium.en—following the methodology commonly adopted in speech editing tasks. Additionally, we measured speaker similarity using the pretrained WavLM-Large model Chen et al. (2022) to compute cosine similarity between reference and synthesized speaker embeddings. We also report the Mel-Cepstral Distortion (MCD) Kubichek (1993) to assess spectral fidelity.

For the subjective evaluation, we randomly selected 80 samples from our test set and generated corresponding speech outputs using VoiceCraft and our proposed model. To ensure a balanced assessment, we conducted a listening study involving 20 native or near-native English speakers (C1–C2 proficiency level). Participants were randomly assigned to two groups of 10, with each group evaluating a total of 120 audio clips: 40 generated by VoiceCraft, 40 produced by MAVE, and 40 ground-truth recordings. Each participant rated the naturalness and intelligibility of the audio samples on a 5-point Likert scale (1 = poor, 5 = excellent). Further details regarding the experimental setup, rating criteria, and instructions provided to participants are available in Appendix A.3.1.

MAVE outperformed VoiceCraft in MOS evaluations across both naturalness (3.48 vs. 3.22) and intelligibility (4.20 vs. 4.01). While the overall gap between our model and the ground truth remains notable, a closer analysis reveals a reduced performance disparity under certain conditions. As shown in Table 3, which breaks down results by the number of words to be generated, with split points selected to ensure approximately balanced sample sizes across ranges, specifically 17, 22, 21, 20 for the different ranges in ascending order. The performance gap narrows significantly in the 8–15 word range. In this interval, our model achieves a naturalness score of 3.71 compared to the ground truth's 3.87, and an intelligibility score of 4.26 versus 4.36, indicating competitive quality for medium-length utterances, while the gap becomes more significant with the increasing length of the target transcript. This trend is consistent with what the model is trained on, as it was trained on the Gigaspeech dataset, which consists of speech with moderate utterance lengths.

## 4.3 EFFICIENCY ANALYSIS

In this section, we compare our model, with previous SOTA AR model for speech editing, namely VoiceCraft Peng et al. (2024), which uses a transformer backbone. All inference experiments were

Table 3: MOS comparison across different transcript word-length ranges.

| Model | Naturalness MOS | | | | Intelligibility MOS | | | |
|---|---|---|---|---|---|---|---|---|
| | 8–15 | 15–22 | 23–34 | >34 | 8–15 | 15–22 | 23–34 | >34 |
| VoiceCraft | $3.44 \pm 0.14$ | $3.17 \pm 0.15$ | $3.17 \pm 0.14$ | $3.11 \pm 0.15$ | $4.07 \pm 0.07$ | $4.00 \pm 0.07$ | $3.95 \pm 0.07$ | $4.01 \pm 0.15$ |
| MAVE (ours) | $\mathbf{3.71} \pm 0.15$ | $\mathbf{3.45} \pm 0.16$ | $\mathbf{3.35} \pm 0.15$ | $\mathbf{3.46} \pm 0.16$ | $\mathbf{4.26} \pm 0.14$ | $\mathbf{4.28} \pm 0.13$ | $\mathbf{4.09} \pm 0.12$ | $\mathbf{4.19} \pm 0.12$ |
| Ground Truth | $3.87 \pm 0.18$ | $3.78 \pm 0.16$ | $3.93 \pm 0.14$ | $4.03 \pm 0.15$ | $4.36 \pm 0.14$ | $4.38 \pm 0.14$ | $4.33 \pm 0.12$ | $4.54 \pm 0.11$ |

Table 4: Performance comparison of MAVE (ours) and VoiceCraft on the RealEdit dataset.

| Model | KV Cache | Inference Time | Tokens / Sec | Avg / Max Mem. (GB) |
|---|---|---|---|---|
| MAVE (ours) | X-attn | 5 min 17 s | 53.5 | **6.2 / 6.5** |
| VoiceCraft | Yes | **4 min 33 s** | **74.9** | 37.9 / 40.2 |
| VoiceCraft | No | 22 min 31 s | 15.1 | 9.2 / 9.7 |

Table 5: Ablation study of model architectures by conditioning mechanism.

| Decoder | Conditioning | WER↓ | MCD↓ | F0↓ | PESQ↑ |
|---|---|---|---|---|---|
| Mamba (ours) | X-Attn. | **7.8** | **4.29** | **0.280** | **2.08** |
| Trm. | X-Attn. | 10.8 | 4.58 | 0.293 | 1.99 |
| Mamba | Concat. | 13.0 | 4.48 | 0.284 | 2.02 |

conducted on NVIDIA A100 GPU. Table 4 reports the inference time required to process the entire RealEdit dataset as well as the average and peak memory consumption. It is important to note that these reported numbers are based on a single generation per sample. The results clearly demonstrate the superior memory efficiency of MAVE compared to VoiceCraft with KV caching, requiring approximately six times less GPU memory—largely due to the selective state mechanism of the Mamba backbone. While VoiceCraft exhibits faster inference speed in this evaluation, this advantage is primarily attributed to the relatively short duration of the target audio segments. On average, our model generates about 85 tokens per sample, corresponding to approximately 1.7 seconds of speech, a regime in which autoregressive overheads are minimal.

However, theoretical complexity analysis reveals that MAVE scales more favorably with sequence length. For longer generation tasks, our model not only maintains significant memory savings but also surpasses VoiceCraft in computational efficiency, offering better speed performance for extended audio synthesis (see A.4 for more details).

### 4.4 ABLATION STUDY

For the ablation study, we considered the masked reconstruction task, in which we used a random subset of the Gigaspeech validation dataset, randomly masked a sequence of words and required the model to reconstruct them (details provided in A.2). Table 5 presents a comprehensive ablation study comparing different architectural designs under similar conditions, including model size (approximately 830M parameters) and training setup. The results highlight that neither a pure transformer-based encoder-decoder architecture nor a standalone Mamba decoder achieves optimal performance, highlighting the importance of our hybrid design.

The encoder-decoder transformer, with the encoder utilized to get contextualized text tokens, cross-attention to condition the audio decoder on text, and transformer decoder to model audio dependencies, achieves moderate performance but lags behind our model in all metrics, particularly in WER (10.8 vs. **7.8**). In contrast, the Mamba-only model, where text and audio tokens are concatenated and processed autoregressively, performs significantly worse with a higher WER (13.0).

Crucially, our MAVE architecture integrates both components, it uses cross-attention to explicitly condition the generation process on textual input while leveraging the Mamba decoder to efficiently model acoustic dependencies. As shown in Table 5, this hybrid approach yields superior performance across all metrics, achieving the lowest WER and MCD, best fundamental frequency (F0) accuracy, and highest PESQ score. These improvements demonstrate that the strength of MAVE does not arise from either cross-attention or Mamba in isolation, but from the combination of both.

## 5 CONCLUSION

In this work we introduce MAVE, a novel hybrid architecture for high-fidelity speech synthesis that combines cross-attention and structured state-space modeling via Mamba within a single autoregressive framework. By leveraging Mamba for robust audio modeling and cross-attention for precise text–audio alignment, MAVE sets a new benchmark for speech editing and zero-shot text-to-speech in terms of both naturalness and efficiency. As future work, we plan to train MAVE on significantly longer audio sequences to further exploit Mamba's ability to capture long-range dependencies.

## 6 ETHICS STATEMENT

The development and deployment of speech editing models like MAVE raise important ethical concerns that must be carefully addressed. One primary issue is the potential for misuse in generating deceptive audio content, such as deepfakes, which could be exploited for misinformation, fraud, or impersonation. While our model enables beneficial applications, such as facilitating editing tasks by correcting errors in speech recordings, improving accessibility, or enhancing voice interfaces.

To mitigate risks, we emphasize the importance of rational use, transparent provenance, and deepfake detection mechanisms. We encourage the integration of watermarking or authentication protocols to help distinguish synthetic speech from authentic recordings. Furthermore, we advocate for clear usage policies and informed consent when applying such technology to the voices of real individuals.

All data used in this study was sourced from publicly available datasets with permissive licenses. No personal or sensitive data were collected or used without proper authorization.

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

# A   APPENDIX

## A.1   IMPLEMENTATION DETAILS

Configuration for our model, and the variants used in ablation studies are shown in Table 6. All models were trained under consistent experimental settings, using Scaled Adam optimizer and Eden Secheduler proposed in Yao et al. (2024), with a base learning rate of 0.01. Hyperparameters were carefully selected to ensure a fair comparison, with all models constrained to have approximately 830 million parameters.

Table 6: Model architecture details for different configurations.

| Model | #Encoder Layers | #Decoder Layers | Model Dimension |
|-------|-----------------|-----------------|-----------------|
| MAVE | 4 | 12 | 1808 |
| Encoder-decoder Transformer | 4 | 12 | 1840 |
| Mamba-only | – | 16 | 2016 |

Table 7 shows detailed configuration used to train the models.

Table 7: Training configuration parameters.

| Parameter | Value |
|-----------|-------|
| Learning rate (lr) | 0.01 |
| Max audio length | 20 |
| Min audio length | 2 |
| Codebook weight | [0.25, 0.25, 0.25, 0.05, 0.05, 0.05, 0.05, 0.05] |
| Number of transformer heads | 16 |
| Type | bfloat16 |
| AMP (Automatic Mixed Precision) | true |

## A.2   ABLATION STUDY SETUP

To conduct ablation studies, we employed the masked reconstruction task. For dataset construction, we used the evaluation subset of GigaSpeech Chen et al. (2021), from which we randomly sampled 500 utterances. We applied several filtering criteria to ensure data quality: (1) utterances with a word error rate (WER) greater than 0.1 when transcribed by Whisper-medium.en Radford et al. (2022b) were excluded to ensure reliable reference transcripts; and (2) utterances containing fewer than five words were discarded to provide sufficient linguistic context.

For each remaining utterance, we mask a contiguous span of $m$ words, where $m$ is sampled uniformly from $[1, M]$, and:

$$M = \min(L - 5, 15),$$

where $L$ the number of words in the transcript of the utterance, ensuring that at least five words remained unmasked to preserve contextual coherence. The starting position of the masked span was selected uniformly at random from all valid positions. After processing, the final dataset consisted of 266 high-quality samples used for ablation analysis.

## A.3   INSTRUCTIONS AND DETAILS FOR USER STUDY

We have conducted a total of 4 user studies. For each of them, we have created a telegram bot to facilitate interaction. In the following subsections, we provide detailed descriptions of the experimental setup, including the instructions presented to users and representative screenshots of the bot interface.

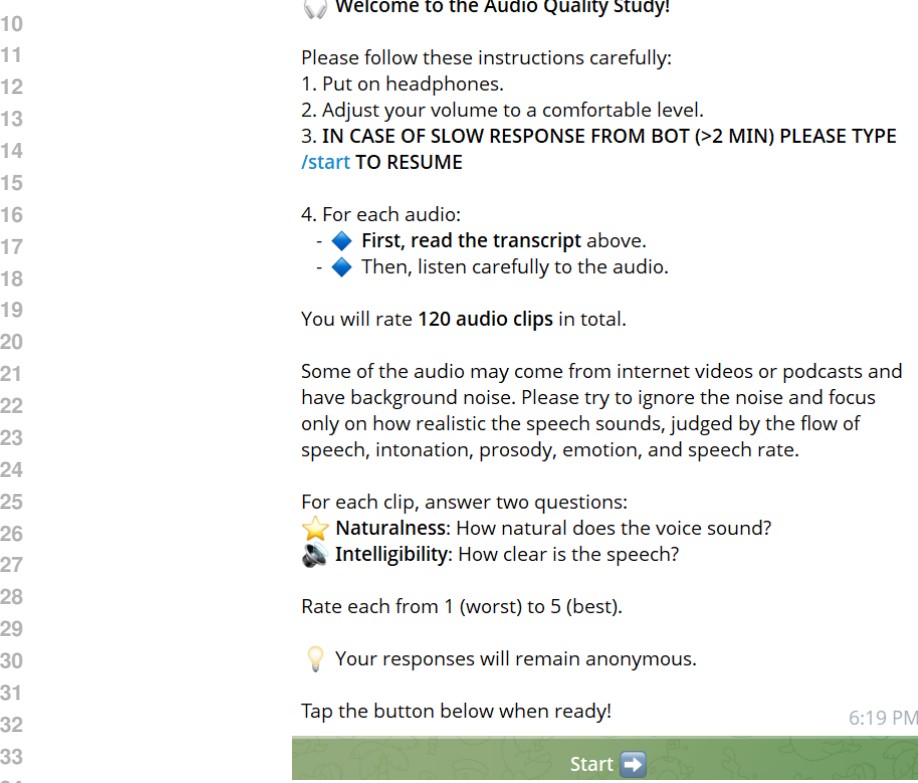

Figure 3: The instructions to assess the quality of different audios in terms of naturalness and intelligibility

### A.3.1 SPEECH EDITING AND ZERO-SHOT TTS MEAN OPINION SCORE

This study was conducted to evaluate the quality of audio samples generated by our model in comparison to VoiceCraft and ground-truth recordings, to provide insights of how well our model performs against state-of-the-art models and to assess how closely our synthesized speech approaches the quality of natural, human speech. We sampled 80 random examples from the RealEdit benchmark to evaluate the models on the speech editing task, and another 80 from the LibriTTS to evaluate on zero-shot TTS. Then, for each task we generated the synthesized audios using both our model MAVE and VoiceCraft. After that we divided these audios into two groups, each group was assigned 120 audios (40 from each category). We then asked 20 people to assess the quality of the generation, assigning each user to one group, ensuring that each group receives exactly 10 people. Audios are randomly shuffled for each new user. At the beginning of the study, the instructions are provided to the users as illustrated in Fig 3. Then for each audio sample, participants were first presented with the corresponding transcript and asked to rate the naturalness of the speech as depicted in Fig. 4. After submitting their naturalness rating, they were then prompted to evaluate the same audio in terms of intelligibility.

### A.3.2 COMPARATIVE STUDIES

To further assess the quality of the generated audios, and to enable direct comparison between the different methods, we have conducted 2 comparative user studies. The first study evaluates our model against VoiceCraft and FluentSpeech. We selected the 14 publicly available audio samples generated by both VoiceCraft and FluentSpeech, which are provided on the VoiceCraft demo page `https://jasonppy.github.io/VoiceCraft_web/`. Using the same text prompts, we generated corresponding outputs using MAVE. This resulted in three pairwise comparisons: (1) VoiceCraft vs. FluentSpeech, (2) VoiceCraft vs. Ours, and (3) FluentSpeech vs. Ours), with 14 pairs per comparison, yielding a total of 42 unique audio pairs.

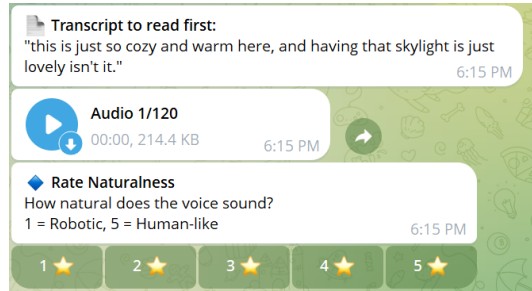

Figure 4: Example of questions users were asked to assess the naturalness for the speech editing task

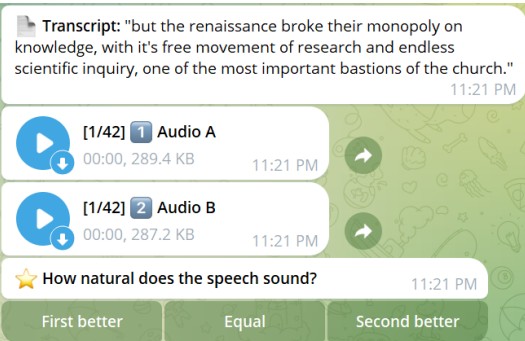

Figure 5: Question example about pair-wise comparative study between our model, VoiceCraft and FluentSpeech

We asked 20 people of proven English fluency to evaluate each pair and choose which is better in terms of both naturalness and intelligibility, or to indicate that they are perceptually equal (three options: "First is better," "Second is better," or "Equal") as show in Fig. 5. To minimize positional bias, the order of the two audios in each pair was randomized across participants, and the presentation order of the pairs themselves was shuffled for each user.

The second comparative study focuses on a direct comparison between MAVE and ground truth audio for the speech editing task. To accomplish this, we have selected 40 random samples from RealEdit data, provided the ground truth before editing and our edited audio, along with their respective transcript. In total, 10 native or near-native English speakers—were asked to compare the two audios in terms of naturalness, and indicate whether the ground truth sounded better, the MAVE-edit sounded better, or there was no perceptual difference between the two. Fig. 6 illustrates the instructions provided to users at the beginning of the study, and Fig. 7 provides an example interface of the side-by-side comparison.

## A.4 Theoretical Analysis of Computational Complexity

In this section, we provide a theoretical comparison between two autoregressive generation paradigms for modeling paired text–audio data. Let $X$ denote the input text sequence of length $L_x$, and $Y$ the output audio sequence of length $L_y$. We study the computational complexity of:

1. a **decoder–only Transformer** that consumes the concatenation of the text and the previously generated audio tokens, and

2. an **encoder–decoder** architecture in which a Transformer encoder processes the text and a stack of Mamba Gu & Dao (2023) layers autoregressively generates the audio. Each Mamba layer is followed by a cross–attention module conditioning on the encoder output.

Throughout the analysis we use the following notation:

🎧 **Naturalness Comparison Study**

Thank you for participating!

You will compare **speech samples** from two systems:
- Your task: judge which sounds more natural.

📌 For each of the **40** pairs:
1. 🔷 Read the transcript.
2. 🔷 Listen to both clips (A and B).
3. Tap: **Audio A, Equal**, or **Audio B**.

💡 Evaluate based on:
• Speech rhythm, emotion, and flow
• Human-likeness and intonation
• Natural pauses and prosody

Figure 6: Instructions provided to users at for pair-wise comparison between our model and ground truth

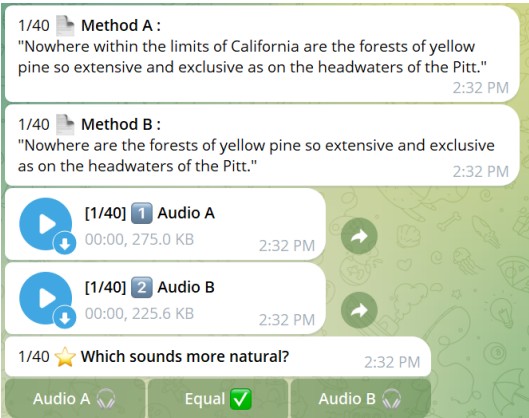

Figure 7: Example of questions asked to user to compare our edited audio with ground truth

- $N_d$: number of layers in the decoder–only Transformer, with hidden dimension $H_0$.

- $N_e$: number of encoder layers in the encoder–decoder model.

- $M_d$: number of Mamba layers in the encoder–decoder model, each with hidden dimension $H$ (assumed equal in encoder and decoder).

Feed–forward expansion factors and head counts are absorbed into the $\mathcal{O}(\cdot)$ notation.

**Decoder–only Transformer.** At generation step $t$ the model processes a sequence of length $L_x + t - 1$. With standard key–value (KV) caching, the cost of computing self–attention for the new token is

$$\mathcal{O}\big(N_d H_0 (L_x + t - 1)\big),$$

while the feed–forward update for the new token adds $\mathcal{O}(N_d H_0)$. Summing over $t = 1, \ldots, L_y$ yields a total complexity

$$\mathcal{O}\big(N_d H_0 \left[L_y L_x + \tfrac{1}{2} L_y^2\right]\big), \tag{4}$$

where the quadratic term $L_y^2$ arises because each new token must attend to all previously generated tokens.

**Encoder–decoder with Mamba decoder.** The text encoder is run once, with cost

$$\mathcal{O}\big(N_e H L_x^2\big), \tag{5}$$

dominated by the self–attention layers. During generation, each Mamba layer updates its hidden state in $\mathcal{O}(H)$ time per token and attends to the fixed encoder output of length $L_x$. Thus the per–token decoder cost is $\mathcal{O}\big(M_d H (L_x + 1)\big)$, and generating the entire audio sequence requires

$$\mathcal{O}(M_d H L_y (L_x + 1)). \tag{6}$$

Unlike the decoder–only case, there is no dependence on $L_y^2$ because the Mamba recurrence does not revisit past audio tokens.

**Comparison.** Equations equation 4–equation 6 highlight a key difference: the decoder–only Transformer grows *quadratically* with the audio length $L_y$, whereas the proposed encoder–decoder with Mamba decoder grows only *linearly* with $L_y$ (after a one–time $\mathcal{O}(N_e H L_x^2)$ encoder cost). When the audio output is much longer than the text prompt ($L_y \gg L_x$), the dominant terms simplify to

Decoder–only: $\mathcal{O}(N_d H_0 L_y^2)$, Encoder–decoder: $\mathcal{O}(M_d H L_x L_y)$.

Because $L_x$ remains fixed while $L_y$ can be very large, the encoder–decoder architecture scales far more favorably for long audio generation.

**Implications for our model.** In our experiments the number of audio tokens is significantly larger than the number of text tokens ($L_y \gg L_x$). Under this regime the proposed encoder–decoder model with a Mamba decoder offers two clear advantages:

1. **Computational efficiency:** linear growth in $L_y$ rather than quadratic, leading to faster autoregressive generation and lower memory usage.

2. **Memory efficiency:** although both models can use KV-cache, the decoder-only Transformer must store keys and values for all past text and audio tokens ($\mathcal{O}(N_d H_0 (L_x + L_y))$ per layer) which grows linearly as the generation goes, whereas the encoder–decoder model only stores a fixed encoder cache ($\mathcal{O}(N_e H L_x)$) and the hidden state of Mamba. When $L_y \gg L_x$, this results in significantly lower memory usage for the encoder–decoder model.

3. **Better conditioning:** the cross–attention mechanism allows the decoder to efficiently incorporate the entire encoded text representation at every step without revisiting the growing audio history.

These properties make the encoder–decoder with Mamba decoder a natural choice for high–fidelity speech synthesis tasks where the output sequence length greatly exceeds that of the input text.

## A.5 USE OF LARGE LANGUAGE MODELS

The authors acknowledge the use of a large language model (LLM) solely for language editing and grammatical refinement of the current manuscript. All scientific content, analysis, and interpretations presented herein are the sole responsibility of the authors.

