# OpenReview forum: "Speak, Edit, Repeat: High-Fidelity Voice Editing and Zero-Shot TTS with Cross-Attentive Mamba"
_ICLR.cc/2026/Conference — Submitted to ICLR 2026_

### Official Review · Reviewer_zYYp · 2025-10-27

**Soundness:** 1
**Presentation:** 2
**Contribution:** 1
**Rating:** 0
**Confidence:** 5

**Summary:**

This paper proposes **MAVE**, a hybrid architecture leveraging a Mamba-based state-space sequence model enhanced with cross-attention for efficient, high-fidelity, and context-aware speech editing and zero-shot text-to-speech (TTS) generation. MAVE models long-range dependencies in audio tokens via Mamba, while dynamically aligning text and audio through cross-attention on phoneme embeddings, enabling bidirectional and precise speech modifications. The empirical evaluation on standard benchmarks demonstrates performance gains in naturalness, intelligibility, and speaker consistency compared to contemporary baselines such as VoiceCraft and FluentSpeech.

**Strengths:**

1. The integration of Mamba SSMs with a cross-attention mechanism tailored for audio-text alignment addresses the quadratic inefficiency of transformer-based decoders in long speech generation.

**Weaknesses:**

1. Lack of novelty. This work is essentially a copy of VoiceCraft in every aspect — design, writing, and overall structure — with the only difference being that the model architecture is changed from Transformer to Mamba.

2. A severe lack of baselines, such as F5-TTS and MaskGCT, which both support TTS and editing.

**Questions:**

N/A

---

> ### Author Response · Authors · 2025-11-21
> **Response to Reviewer zYYp**
>
> $\textbf{Q-1: ``Lack of novelty. ..."}$
>
> $\textbf{R-1:}$ We thank the reviewer for his feedback but we strongly disagree with his overall assessment of our work. The reviewer's assertion is that our work is a mere replication of VoiceCraft. While we acknowledge VoiceCraft as a leading prior method and thus adopt its evaluation protocol to ensure a fair and meaningful comparison, our technical contributions are distinct and novel in several key aspects as have been clearly recognized by the rest of the reviewers.
>
> To further clarify this, we list below the core contributions of our work and explain how MAVE differs from VoiceCraft.
>
> First, in MAVE we introduce a **novel** hybrid encoder-decoder framework that fundamentally differs from VoiceCraft's decoder-only Transformer-based design in two key aspects:
>
> - **Audio Modeling:** We replace the Transformer with a selective State Space Model (Mamba) in the audio pathway, enabling efficient, linear-complexity modeling of long audio sequences while preserving speaker characteristics.
>
> - **Text-audio Conditioning:** Rather than concatenating text and audio tokens (as in VoiceCraft), we employ a dedicated Transformer-based text encoder and condition the Mamba audio decoder via cross-attention. This decouples modality processing and allows for more precise alignment between linguistic content and acoustic realization.
>
> These architectural changes are crucial, since as we have demonstrated in our ablation studies (see Table 5) a naïve “mere replacement” of Transformer with Mamba performs poorly (the worst out of all possible combinations). This clearly demonstrates that our gains stem not from a trivial architecture modification, but from a carefully engineered integration of SSMs into the speech editing pipeline.
>
> Moreover, MAVE achieves state-of-the-art results in speech editing while using 6× less memory during inference and generating outputs that are perceptually closer to the ground truth audio.
>
> Another critical differentiator is MAVE's significantly enhanced robustness and output stability compared to VoiceCraft. VoiceCraft necessitates specialized post-processing and engineering tricks to achieve acceptable output quality; specifically, it requires $N$ multiple sampling runs to mitigate a tendency for generating prolonged silent segments, along with explicit detection and probability reduction of silence tokens. This essential requirement effectively multiplies VoiceCraft's inference time by a factor of $N$. In contrast, MAVE inherently produces high-fidelity, silence-free output in a single forward pass. We achieve this robust performance without any need for external post-processing steps or custom output engineering, which yields a decisive end-to-end speed advantage in real-world use cases.
>
> Once again, we have followed VoiceCraft’s evaluation protocol only where necessary for fairness, not out of imitation. If the reviewer believes there are specific instances of textual or structural overlap beyond standard academic framing, we welcome concrete examples and we are committed to fully address them. However, we firmly maintain our position that MAVE presents a novel and technically distinct contribution to efficient, high-quality speech editing.
>
> $\textbf{Q-2: ``A severe lack of baselines, such as F5-TTS and MaskGCT, which both support TTS and editing."}$
>
> $\textbf{R-2:}$ We agree that there are recent works that need to be included in our comparisons. We have included F5-TTS and MaskGCT that are suggested by this and other reviewers in our comparisons, which are provided in the "General Response" (Part-A) section. We believe that our response in that section fully addresses the reviewer's concerns.

---

> > ### Comment · Area_Chair_7Z5x · 2025-11-28
> > **Please engage in the discussion**
> >
> > Dear Reviewer,
> >
> > The authors have provided a response to your review. As your original review was highly critical of the novelty and thoroughness of the evaluation but provided limited detail, could you please engage in the discussion to expand on your thoughts? It is very important to provide more context to justify your ratings and assist us in reaching a final decision.
> >
> > Best regards,
> > AC

---

### Official Review · Reviewer_niSb · 2025-11-01

**Soundness:** 3
**Presentation:** 3
**Contribution:** 3
**Rating:** 6
**Confidence:** 4

**Summary:**

The work proposes the first structured-state-space model (Mamba) successfully adapted for text-conditional speech generation (Voice Editing and Zero-Shot TTS) by fusing it with cross-attention layers for linguistic alignment. The proposed approach outperforms autoregressive and diffusion-based approaches in fidelity, efficiency, and robustness.

**Strengths:**

- The paper is novel and will intrigue the community since it is the first paper that uses MAMBA for these speech tasks according to the authors. I do remember that some people used it in the past but for different tasks (Mamba in Speech: Towards an Alternative to Self-Attention).
- The paper is very well presented with very good experiments and explanations. Good presentation of both objective and subjective metrics. I do like the very well set up subjective evaluation and the details provided in the Appendix.

**Weaknesses:**

- The only weakness that I see is that the authors did not do a comparison with more Speech Editing or Zero-shot methods. They compared only with VoiceCraft. I would suggest to compare with other models too in order to have a more complete analysis.

**Questions:**

- Interesting masking with a mask token. I haven't seen that before. When I was doing research in Speech Editing we used to use masks of zeros on the mel-spectrogram but now since you use tokens it has to change. Very interesting detail.

- Last paragraph of 3.2.1, can you elaborate a bit more of why MAMBA performs that good? Is there an ablation study for that? This paper will create a lot of discussion in the speech field since the audience is divided on the attention vs SSM, so it would be nice to give explanations on why this works better.

---

> ### Author Response · Authors · 2025-11-21
> **Response to Reviewer niSb**
>
> We thank the reviewer for his feedback and positive assessment of our work.
>
> $\textbf{Q-1 ``The only weakness that ... complete analysis."}$
>
> $\textbf{R-1:}$ The reviewer raises a valid point, which is also shared among other reviewers. As detailed in our General Response (Part-A), we have included additional comparisons with F5-TTS and MaskGCT in Tables R1–R3. The reported results provide further context for MAVE’s performance and help strengthen our evaluation, while fully supporting our original claims.
>
> $\textbf{Q-2 ``Last paragraph of 3.2.1, ... this works better."}$
>
> $\textbf{R-2:}$ We thank the reviewer for his question, which mainly stems from the limited adoption to date of state space models (SSMs) in audio generation, despite their growing use in other sequence modeling domains.
>
> To assess the contribution of Mamba in our overall model, we did conduct an ablation study. As shown in Table 5 (2nd row) of the main paper, we replaced Mamba blocks with Transformer layers of equivalent capacity in our audio pathway, while keeping all other components, including the text encoder and training protocol, identical. We refer to Table 6 (Appendix A.1) for more detailed configurations. From these comparisons, it is clear that our Mamba-based variant consistently outperforms the Transformer counterpart across all evaluation metrics, confirming the performance gain of the SSM architecture.
>
> Furthermore, we would like to highlight that our architectural decision to integrate the Mamba state-space model (SSM) exclusively within the audio pathway is primarily driven by its inherent strengths in efficiently modeling long-range high-level temporal dependencies. This characteristic is particularly well-suited for audio processing, where the task often relies on capturing high-level, global temporal patterns rather than the fine-grained, token-by-token dependencies crucial in other modalities.
> In contrast, we used transformer for text encoder rather than Mamba, because text encoding mandates a highly detailed semantic understanding and robust bidirectional context integration for exact text-audio alignment. Prior work has indicated that the standard Mamba architecture exhibits sub-optimal performance relative to the Transformer model when applied to tasks requiring this level of precise semantic parsing. We refer to [41] for a detailed analysis of Mamba's limitations in this domain.

---

### Official Review · Reviewer_zZAj · 2025-11-01

**Soundness:** 3
**Presentation:** 3
**Contribution:** 3
**Rating:** 6
**Confidence:** 2

**Summary:**

This paper introduces MAVE (Mamba with Cross-Attention for Voice Editing and Synthesis), a novel autoregressive architecture for text-conditioned speech generation. The core innovation is a hybrid design combining a Mamba (SSM) backbone for efficient, long-sequence audio modeling with a cross-attention mechanism for robust text conditioning. This architecture is designed to effectively manage the significant length mismatch between text and audio modalities.

The authors evaluate the model on two primary tasks. For speech editing, MAVE achieves performance on the RealEdit benchmark that is **comparable** to the state-of-the-art VoiceCraft model, with results largely within the confidence intervals of the baseline. The model's strengths are more clearly demonstrated in zero-shot text-to-speech (TTS), where MAVE achieves **statistically significant improvements** over VoiceCraft in both MOS naturalness and intelligibility.

A key contribution of this work is its computational efficiency; MAVE requires approximately **6x less inference memory** than the Transformer-based VoiceCraft. The strong zero-shot TTS performance is presented as a direct application of the model's ability to learn speaker characteristics from audio context, a mechanism that is learned during its training on the speech editing (in-filling) task.

**Strengths:**

*   **Originality:** The paper's primary original contribution is the MAVE architecture. It proposes a hybrid design that combines a Mamba backbone, chosen for its linear-time efficiency in modeling long audio sequences, with a separate cross-attention mechanism for text conditioning. This architecture is a well-motivated solution for applying SSMs to a cross-modal task, specifically addressing the significant length mismatch between text and audio sequences. This integration of a Mamba decoder with a flexible, length-agnostic cross-attention module is a novel approach in this domain.
*   **Clarity:** The paper is clearly written. The core architectural idea is presented logically and is supported by Figure 1, which effectively visualizes the text/audio data flow and the detailed decoder block. The authors explain the token rearrangement strategy that unifies the speech editing and TTS tasks under a single autoregressive framework. The inclusion of a theoretical complexity analysis in the appendix (A.4) also adds to the clarity of the model's proposed benefits.
*   **Quality:** The paper's claims are supported by a thorough evaluation. On the primary speech editing task (Table 1), the MOS scores for naturalness and intelligibility are comparable to the VoiceCraft baseline, with results largely falling within the reported confidence intervals. While this demonstrates competitive performance, the model's quantitative strengths are more clearly shown in the zero-shot TTS evaluation (Table 2). Here, the model achieves a statistically significant improvement over the baseline in both naturalness (3.48 vs. 3.22) and intelligibility (4.20 vs. 4.01). Furthermore, the ablation study (Table 5) is of high quality and provides a strong justification for the Mamba + Cross-Attention design, showing it outperforms both a "Mamba-only" (concat) approach and a "Transformer + Cross-Attention" model.
*   **Significance:** The significance of this work is twofold. First, from a practical standpoint, the paper presents a model that achieves competitive-to-superior generative quality while being significantly more efficient. The reported \~6x reduction in inference memory (Table 4) is a significant practical contribution, potentially making SOTA-level speech generation more accessible. Second, from a scientific standpoint, this work provides a viable blueprint for replacing the dominant Transformer backbone in complex, text-conditioned autoregressive audio models. It demonstrates that SSM-based hybrids can offer a favorable trade-off between performance and efficiency, which may encourage further research into similar architectures.

**Weaknesses:**

The claim in the abstract that the model is "not explicitly trained on the \[zero-shot TTS] task" is potentially misleading. Section 3.2.3 clarifies that the editing task uses surrounding audio for speaker context, while the TTS task uses a prepended prompt. The core mechanism—conditioning on audio tokens for speaker identity—appears to be a fundamental part of the training objective, not a purely emergent capability. A more precise framing would be that zero-shot TTS is a direct and successful *application* of the speaker context mechanism learned during in-filling. The authors are encouraged to clarify this framing in the final version.

The paper's claims about state-of-the-art *speech editing* performance are not strongly supported by the data in Table 1. The MOS scores for MAVE versus VoiceCraft on the RealEdit benchmark are very close, with overlapping confidence intervals, suggesting performance is, at best, on par. This SOTA claim is further weakened, as the authors note, by the evaluation being on an incomplete version of the RealEdit benchmark. In contrast, the model's superiority is much clearer in the zero-shot TTS task (Table 2) and its efficiency (Table 4). The paper would be stronger if it re-framed its primary contribution around these more significant and clearly demonstrated achievements: namely, achieving *comparable* editing quality and *superior* TTS quality with a *dramatically* more efficient architecture.

A significant practical limitation of the "speech editing" framework is its reliance on manual segmentation. The model requires the user to explicitly define the "before" and "after" audio spans for an edit. It cannot, for example, take a full audio file and a corrected transcript and "automatically find and fix" the errors. This makes it a powerful *component* for an editing tool, but not a fully automatic "corrector," which limits its immediate practical utility. This limitation should be discussed, and a constructive path for future work would be to investigate integrating this model with an automatic text-audio aligner to create a true, end-to-end "find-and-fix" system.

The zero-shot TTS evaluation, while showing strong results on LibriTTS, could be made more robust. First, the evaluation is on clean read-aloud speech, which does not directly test the model's main strength: its training on "in-the-wild" Gigaspeech audio. Second, Table 3 shows a clear trend of the performance gap to ground truth widening as the generated text gets longer. To strengthen the paper's claims, the authors could (1) add a TTS evaluation on an "in-the-wild" test set (e.g., held-out Gigaspeech samples) to validate its robustness, and (2) provide a brief analysis of *why* long-form quality degrades (e.g., is it speaker similarity or text alignment?) to better guide future work on Mamba's long-context state.

**Questions:**

*   Could the authors please clarify the "not explicitly trained for zero-shot TTS" claim? Section 3.2.3 implies that speaker context is learned from surrounding audio tokens during editing. How does this mechanism fundamentally differ from prepending a reference prompt for zero-shot TTS, which seems like a direct application of the same learned capability?
*   The practical utility of the editing feature relies on manual segmentation of the "before" and "after" audio. Have the authors investigated a path to a fully automated system, for instance, by combining MAVE with a text-audio aligner that could automatically identify and propose mismatched spans for correction?
*   Given the paper's focus on Mamba's efficiency, did the authors experiment with replacing the Transformer-based text encoder with a Mamba-based one? This could create a more architecturally homogenous model and potentially yield further efficiency gains.
*   Section 3.2.3 mentions "cross-speaker editing" as a possibility. Was this capability evaluated? For example, how well can the model edit a phrase into a target speaker's voice using a reference prompt from a *different* speaker?

**Details Of Ethics Concerns:**

The primary ethical concern is the dual-use nature of this technology, which falls under `potential harmful insights, method & application`. The MAVE model is a highly effective tool for generating audio deepfakes. Its zero-shot capability significantly lowers the barrier for misuse, requiring only a few seconds of a target's voice to generate new, arbitrary speech. The paper's own results show this synthetic speech is often indistinguishable from the original, making it a potent tool for misinformation, such as faking statements from public figures, or for fraud.

These risks directly impact `privacy, security, and safety`. The method enables severe violations of an individual's voice privacy by allowing non-consensual cloning. This, in turn, poses a security risk, as the technology could be used to bypass voice-based authentication systems. Such applications threaten personal and financial safety through impersonation scams, extortion, or targeted harassment.

While the authors acknowledge these risks in their ethics statement, the mitigations they propose are societal suggestions rather than technical safeguards. Given the model's high performance, robustness, and potential for harm, this paper warrants a specialized review by the ethics committee to fully assess its societal impact.

---

> ### Author Response · Authors · 2025-11-21
> **Response to Reviewer zZAj (Part-A)**
>
> $\textbf{Q-1: ``Could the authors clarify ... same learned capability?"}$
>
> $\textbf{R-1:}$ We thank the reviewer for raising this point and giving us the opportunity to clarify the intended meaning behind our statement. Our intention was to highlight that the training strategy of MAVE focused on specifically for speech editing — not zero-shot TTS. Had TTS been the objective, the architecture and training protocol would require fundamental modifications such as:
>
> 1. Span structure: Editing demands multi-span infilling (up to 3 disconnected spans in audio), whereas TTS generates single continuous span over longer utterances.
>
> 2. Editing is inherently non-autoregressive: employed token rearrangement mechanism (Sec 3.1) preserves context coherence across edit boundaries, necessitating token shuffling — a technique redundant for autoregressive TTS.
>
> 3. Editing uniquely requires prosodic continuity between pre-edit and post-edit segments (e.g., pitch/rhythm alignment), a constraint absent in TTS where prompts and targets are independent utterances (e.g. VoiceStar).
>
> 4. TTS systems typically train on and generate long-form audio sequences, whereas speech editing operates on constrained segments; accordingly, MAVE trains exclusively on short clips ($\leq$ 20 seconds) to maintain boundary coherence.
>
> In the updated manuscript, we will make this distinction between speech editing and TTS-focused training clear.
>
> Please note that retraining MAVE for TTS would entail discarding its core editing-specific components: multi-span handling and token rearrangement. Nevertheless, MAVE can accommodate this TTS-oriented training pipeline which could potentially lead to improved performance. Unfortunately, due to the limited time and resources we had in our disposal for the preparation of this rebuttal we were unable to perform such model training. However, it is an interesting direction we plan to follow in the future. Finally, we want to emphasize once more that, as it is already mentioned in our original manuscript, TTS was not the main focus of this work but a byproduct and we only claim competitive TTS performance and not SOTA with the main focus being speech editing.
>
> $\textbf{Q-2: ``The paper's claims about state-of-the-art speech ... at best, on par."}$
>
> $\textbf{R-2:}$ We thank the reviewer for his observation. The reviewer is correct that the confidence intervals of the MOS scores for MAVE and VoiceCraft on the RealEdit benchmark overlap. However, the overlap occurs only in the outer regions of the ±1σ intervals, while the point estimates consistently favor MAVE (3.90 vs 3.77). Using a similar reasoning, we observe that MAVE overlaps with the ground-truth audio quality, implying that our model achieves a level of perceptual quality that is, for all intents and purposes of this test, indistinguishable from human speech.
>
> In any case, to provide a more definitive evaluation, in our original submission, we conducted in a side-by-side subjective listening test (see Fig. 2, main paper). Participants directly compared MAVE against VoiceCraft, FluentSpeech, and ground-truth audio. These results demonstrate a statistically significant preference for MAVE over both baseline systems. Furthermore, we have also conducted side-by-side comparison between MAVE and ground-truth, in which MAVE was rated as perceptually indistinguishable from the ground truth in a majority of cases. This provides direct and compelling perceptual evidence for MAVE's state-of-the-art performance in speech editing.
>
>
> $\textbf{Q-3: ``This SOTA claim is further weakened, as the authors note, by the evaluation being on an incomplete version of the RealEdit benchmark."}$
>
> $\textbf{R-3:}$ We would like to clarify that our evaluation on a subset of the RealEdit benchmark was necessitated by external constraints beyond our control. Subsequent to the RealEdit benchmark's publication, the Spotify Podcast dataset was retracted by its creators, making full replication infeasible.
>
> Our study therefore uses the largest available and reproducible subset, comprising the LibriTTS and YouTube (Gigaspeech) portions. We have explicitly noted this limitation in the manuscript to ensure transparency, and we believe our evaluation on the remaining, substantial portion of the benchmark provides a rigorous and fair comparison.
>
> **Part-B of our response follows**

---

> ### Author Response · Authors · 2025-11-21
> **Response to Reviewer zZAj (Part-B)**
>
> $\textbf{Q-4: ``The practical utility ... correction?''}$
>
> $\textbf{R-4:}$ We agree with the reviewer that a fully automatic "find-and-fix" system is essential for practical deployment. It is also true that in our paper we do not discuss other parts essential for the automatic speech editing pipeline. Nevertheless, the rest of essential components for automatic system already exist. Specifically, our overall system consists of the following components:
>
> 1. $\textbf{ASR and Alignment}$: Whisper and MFA-Aligner for precise transcription and timing.
>
> 2. $\textbf{Edit Detection:}$ The "Myers diff" algorithm [M] to automatically locate insertions, substitutions, and deletions.
>
> 3. $\textbf{Editing/Synthesis:}$ Our proposed MAVE model to perform the high-quality speech synthesis.
>
> Furthermore, our evaluation on the RealEdit benchmark already utilizes part of this pipeline (the MFA-Aligner), confirming its practical utility.
>
> [M] Myers, E. W. (1986). An O(ND) difference algorithm and its variations. Algorithmica, 1(1-4), 251-266. https://doi.org/10.1007/BF01840446
>
> $\textbf{Q-5: ``Given the paper's focus on Mamba's ... efficiency gains.''}$
>
> $\textbf{R-5:}$ Replacing the Transformer-based text encoder with a Mamba-based encoder is an interesting direction that indeed deserves investigation. Our primary rationale for adopting Mamba in the audio pathway stems from its strength in capturing high-level temporal dependencies efficiently, which aligns well with modeling audio where fine-grained token-by-token reasoning is less critical. In contrast, text encoding typically requires precise, detailed semantic parsing and strong bidirectional context integration, capabilities where Mamba has been shown to underperform relative to Transformers, as noted in prior work (we refer to the work in [41] that discusses this downside of Mamba).
>
> Our architectural decision is strategically motivated by the functional distinction between the encoder and decoder. The decoder's role is to perform unidirectional, autoregressive generation, a process that aligns perfectly with the core operational mode of a standard Mamba state-space block. Conversely, the encoder is tasked with achieving comprehensive full-sequence understanding and bidirectional context integration, a requirement for which the standard unidirectional Mamba is ill-suited. Critically, the decoder operates in autoregressive fashion, making it the primary bottleneck for inference latency. Consequently, optimizing this specific component with Mamba yields a substantially greater impact on overall efficiency and speedup than optimizing the encoder, which processes the input sequence only once.
>
> In summary, we greatly appreciate the suggestion regarding a bidirectional Mamba encoder. We agree this represents a valid and intriguing avenue for future architectural exploration. Nevertheless, our current priority focused on the decoder, as its auto-regressive nature allowed our Mamba implementation to yield the most substantial and impactful performance gains in inference efficiency.
>
> **Part-C of our response follows**

---

> > ### Author Response · Authors · 2025-11-21
> > **Response to Reviewer zZAj (Part-C)**
> >
> > $\textbf{Q-6: ``The zero-shot TTS evaluation ... long-context state.''}$
> >
> > $\textbf{R-6:}$ Per the reviewer’s suggestion, we evaluated MAVE on the noisy subset of LibriTTS (dev-other) to assess robustness under non-ideal conditions. Results for this evaluation are provided in our General Response (see Table R3) and the clean subset (dev-clean; Table R2). A detailed analysis of these outcomes is provided below:
> >
> > 1. While MAVE ranks a very close second in WER, it achieves the highest scores in both UTMOS (naturalness) and speaker similarity
> >
> > 2. We observe that non-autoregressive models (F5-TTS, MaskGCT) can sound more robotic on this challenging data. This aligns with prior work suggesting that autoregressive models are better at preserving prosody and fine-grained acoustic details, which is critical for naturalness (we refer to the work in [33] that discusses this issue).
> >
> > 3. The performance gap in UTMOS is more pronounced in speech editing than in TTS. We hypothesize that editing imposes a stricter requirement: the generated segment must not only be natural in isolation but must also achieve holistic coherence, seamlessly integrating with the surrounding context in rhythm, pitch, and speaker characteristics. MAVE's architecture, with its in-context learning and autoregressive decoding, appears uniquely suited to this challenge.
> >
> > The results on the clean LibriTTS (dev.clean) subset (see Table R2) corroborate the findings from the noisy condition (see Table R3), demonstrating consistent model behavior across different data qualities.
> >
> > Regarding the long-form degradation, we observe that the primary gap between generated audio and ground truth lies in MOS naturalness, while MOS intelligibility remains relatively high. This suggests that words are generally pronounced correctly, but the overall prosody and fluency degrade. Upon listening, we notice that speech pace tends to increase in longer utterances. While this is currently an empirical observation, further analysis is needed to determine whether the issue stems from model architecture or training data limitations. Due to time constraints for the rebuttal, we were unable to conduct a detailed ablation study. Nevertheless, we sincerely thank the reviewer for highlighting this important point, which we agree that represents a promising direction for future work.
> >
> > $\textbf{Q-7: ``Section 3.2.3 mentions "cross-speaker editing ... different speaker?''}$
> >
> > $\textbf{R-7:}$ We agree with the reviewer that an explicit evaluation of cross-speaker editing was not explored in the current work. Our primary focus was on establishing a new state-of-the-art for in-speaker editing, which is the core task addressed by the RealEdit benchmark and the main focus of comparable works.
> >
> > However, the architectural capability for cross-speaker editing is a direct consequence of our model's design. As described in Section 3.2.3, MAVE does not use explicit speaker embeddings but instead relies on in-context learning from reference audio tokens. This is the identical mechanism used for our evaluated zero-shot TTS task. Therefore, in a cross-speaker editing scenario:
> >
> > 1. The "reference utterance" would be a prompt from the target speaker.
> >
> > 2. The "masked audio context" would be drawn from the source audio that needs to be edited.
> >
> > The model would then generate the edited segment in the target speaker's voice, conditioned on the reference prompt, while maintaining the linguistic and prosodic structure of the edit. While we have not formally evaluated this, it is a direct application of the same in-context learning principle demonstrated for TTS. We agree with the reviewer that this is an important capability to clarify. We will revise the manuscript and explicitly state that cross-speaker editing is a compelling direction for future work.

---

### Official Review · Reviewer_CPTT · 2025-11-08

**Soundness:** 2
**Presentation:** 4
**Contribution:** 3
**Rating:** 4
**Confidence:** 4

**Summary:**

This paper introduces MAVE, an autoregressive TTS model that integrates a Mamba backbone (for efficiency) with cross-attention (for text-speech alignment). The inspiration comes from the fact that existing methods are often either high-fidelity but expensive, like Transformers, or efficient but struggle with coherence, like diffusion models . The paper demonstrates that MAVE achieves state-of-the-art performance in speech editing on the RealEdit benchmark as compared to VoiceCraft with 6x less inference memory. It also outperforms VoiceCraft in speaker similarity and naturalness for zero-shot TTS.

**Strengths:**

1. The main contribution of this paper is integrating Mamba with cross-attention to allow conditioning the model on text without explicit alignments. While I am not entirely familiar with related work in this area, this seems to be one of the first papers to do this and is a strong contribution. The ablations in Table 5 support the usefulness of MAVE having both Mamba and cross-attention; Mamba-only and Transformer-only underperform MAVE.
2. The human evaluations show that the model is essentially perceptually equal or better than the ground-truth speech, which is encouraging.

**Weaknesses:**

1. MAVE is only compared to Voicecraft (over all test examples) and FluentSpeech (over a 14-example subset). There are lots of new open-source TTS models, many over a year old; F5-TTS, MaskGCT, VoiceStar. The paper lacks comparisons to a lot of these models and without these, the paper’s claim of state-of-the-art results ‘outperforming leading autoregressive and diffusion models’ is misleading.
2. The authors emphasize one of MAVE’s main advantages is its linear-time complexity. However, the results in Table 4 show that MAVE is actually slower than VoiceCraft on the RealEdit benchmark. The claim of superior speed for longer sequences is purely theoretical (discussed in Appendix A.4)  and is not validated with an experiment. I’d recommend the authors experiment with longer text generations and show that the model is much faster than baselines experimentally.

**Questions:**

1. Can you provide a plot (e.g., sequence length vs. inference time) that shows the crossover point where MAVE actually results in faster wall-clock speed than theTransformer's quadratic scaling?
2. The model’s naturalness reduces as the length of the generation increases (Table 3), as expected. You attribute this problem to the training dataset, which has examples of moderate length. However, it is also possible that the model architecture cannot maintain long-range coherence (although theoretically, given that it is based on Mamba plus cross-attention, it should). Can you design and run an experiment that disentangles these two possible causes?

---

> ### Author Response · Authors · 2025-11-21
> **Response to Reviewer CPTT**
>
> We thank the reviewer $\textbf{CPTT}$ for his valuable critiques, which prompted a deeper investigation into our model's robustness. We believe that the additional analysis and ablation studies undertaken in response to his comments have improved the scope and clarity of our work.
>
> $\textbf{Q-1: ``MAVE is only compared to VoiceCraft ... is misleading?"}$
>
> $\textbf{R-1:}$ We thank the reviewer for pointing out these recent works as they indeed are very relevant. Following the reviewer's suggestion we included comparison with these models, which are detailed in our general response (please see Tables R1-R3).
>
> Regarding VoiceStar, we note that it is a pure TTS model and unfortunately does not support speech editing mode out of the box. Consequently, we are unable to include it in our quantitative comparisons.
>
>
> $\textbf{Q-2: ``The authors emphasize ... RealEdit benchmark?"}$
>
> $\textbf{R-2:}$ We thank the reviewer for raising this point. We would like to clarify that our model's inference speed is not slower than VoiceCraft. We acknowledge a potential source of confusion in Table 4, where the reported execution time for VoiceCraft on the $\text{RealEdit}$ dataset represents a single-sample inference. However, in all our comparisons, we run VoiceCraft for $N$ inference passes (according to the guidelines by the authors of VoiceCraft), which are necessary to mitigate the tendency of the model to generate outputs with long silences. We have failed to explicitly mention this important detail in the main paper, which led to valid confusion. We will make sure to highlight this issue in the updated manuscript.
>
>
> $\textbf{Q-3: ``Can you provide a plot ... quadratic scaling?"}$
>
> $\textbf{R-3:}$ The reviewer has raised a valid concern about validating experimentally the theoretical linear complexity of MAVE. This is addressed in detail in our General Response (Part-B)
>
>
> $\textbf{Q-4: ``The model`s naturalness ... disentangles these two possible causes?"}$
>
> $\textbf{R-4:}$ We agree with the Reviewer that the investigation of whether the drop in naturalness with longer generations stems from architectural limitations or training data constraints can provide helpful insights. Unfortunately, due to time and resource constraints while preparing our rebuttal, we were unable to conduct additional training experiments that could shine some light to this direction.
>
> That said, we can offer two observations that provide indirect but meaningful evidence:
>
> 1. Architecture comparison under identical data constraints: In our current setup, trained exclusively on moderate-length utterances from LibriTTS, MAVE consistently outperforms  Mamba-only based baselines (see Table 5 in Sec-4.4). This suggests that, even when limited by training sequence length, our hybrid Mamba–Transformer architecture better preserves naturalness over extended outputs than Mamba-only. If the degradation were purely architectural, we would not expect MAVE to outperform Mamba under the same training data regime.
>
> 2. Robustness in TTS with increasing length: In zero-shot TTS experiments MAVE shows slower degradation in MOS scores compared to VoiceCraft as utterance length increases. This indicates that our architecture may be better suited to handle longer sequences, possibly due to Mamba’s recurrence-based state propagation, along with explicit text-conditioning via cross attention.
>
> We fully acknowledge that these are indicative rather than conclusive evidence. A definitive test would require training on datasets with systematically varied maximum lengths—a direction we will actively pursue in a follow-up work. We appreciate the reviewer’s suggestion and agree it is a valuable avenue for future investigation.

---

### Author Response · Authors · 2025-11-21
**General Response (Part - A)**

We thank all reviewers for their thoughtful feedback and constructive comments. Few key issues were highlighted by multiple reviewers. In particular, questions were raised about:

(a) Limited comparison with recent models apart from VoiceCraft and FluentSpeech

(b) Validation of the theoretical linear complexity of our proposed MAVE

We provide a unified response to these points here, before addressing individual comments in subsequent sections.

$\textbf{Limited comparison with recent models (F5-TTS, MaskGCT):}$

We fully agree that a comparison with more recent speech editing models is essential. Below we provide results for F5-TTS and MaskGCT and we refer to them in detail in our response below:

**Table R1.** Comparison of speech editing models on RealEdit benchmark

| Model                 | WER (medium) ↓ | WER (large) ↓ | UTMOS ↑ |
|--------------------|--------------------|---------------|---------|
| F5-TTS               | 9.5                      | 11.1             | 3.11    |
| MaskGCT           | 8.6                      | 10.5             | 3.11    |
| VoiceCraft           | 6.9                      | 8.4              | 3.45    |
| **MAVE (ours)**   | **5.9**                | **7.5**          | **3.74**|
| Ground Truth      | 5.2                      | 6.8               | 3.92    |


We would like to clarify that the reason these methods were not included in our initial comparisons is that they are primarily presented as zero-shot text-to-speech (TTS) systems, and neither provides quantitative results nor dedicated benchmarks for speech editing in their original publications. That said, MaskGCT briefly notes on its paper that it can support speech editing and includes two illustrative examples on their webpage. On the other hand, the authors of F5-TTS do not mention speech editing capabilities in their paper, but their GitHub repository contains code demonstrating how to adapt the model for this task. So, we adjusted the official codebases of F5-TTS and MaskGCT for the speech editing and evaluated on RealEdit benchmark. To ensure a fair comparison, all methods utilized the Montreal-Forced-Aligner (MFA) [A] for precise text-audio alignment. The results (see Table R1) demonstrate that MAVE maintains a significant performance advantage over these baselines, achieving the lowest Word Error Rate (WER) and the highest perceptual quality (UTMOS) [B].

Furthermore, we have evaluated MaskGCT and F5-TTS for the TTS task on LibriTTS (dev.clean) with more clean audios (see Table R2). Also, in direct response to reviewer feedback on robustness, we have conducted an additional evaluation on the noisier dev.other split (see Table R3). Our method yields the best scores in speaker similarity (SIM) and perceptual quality (UTMOS) for both noisy and clean subsets.

It is important to underscore that our paper's contribution is a state-of-the-art speech editing model. Its ability to yield $\textbf{``very competitive"}$ zero-shot TTS results (as mentioned in the main paper) is presented as a strong secondary demonstration of its general in-context learning capabilities, not as a new SOTA for the TTS task itself.

**Table R2.** Comparison of models on LibriTTS **clean** (dev.clean) subset
| Model                | WER (medium) ↓ | WER (large) ↓ | SIM ↑ | UTMOS ↑ |
|:------------------:|:--------------:|:-------------:|:-----:|:-------:|
| F5-TTS              | **5.8**               | **6.9**               | 0.56  | 3.71    |
| MaskGCT          | 7.7                | 9.1               | 0.57  | 3.75    |
| VoiceCraft          | 7.5               | 9.3                | 0.55  | 3.93    |
| **MAVE (ours)**  | 6.6          | 7.4           | **0.57** | **4.03** |

**Table R3.**  Comparison of models on LibriTTS **noisy** (dev.other) subset. Ground Truth (GT) is shown separately for reference.
| Model                 | WER (medium) ↓ | WER (large) ↓ | SIM ↑ | UTMOS ↑ |
|:------------------:|:------------------:|:-------------:|:-----:|:-------:|
| F5-TTS               | **6.6**        | **8.6**       | 0.52      | 3.50      |
| MaskGCT           | 8.6             | 10.1          | 0.49      | 3.37      |
| VoiceCraft          | 8.1             | 9.8             | 0.54      | 3.54      |
| **MAVE (ours)**  | 6.8             | 8.9             | **0.56** | **3.68** |
| GT                      | 3.8             | 4.9             | 0.64      | 3.82      |

---

> ### Author Response · Authors · 2025-11-21
> **General Response (Part - B)**
>
> $\textbf{Experimental Validation of Linear Complexity:}$ The reviewers make another valid point and we thank them for the opportunity to clarify this. We constructed a benchmark where audio length varies from 2 to 20 seconds. For each length, we perform 10 editing operations (randomly masking up to 70\% of content) and report the average inference time and peak GPU memory for MAVE and VoiceCraft (with and without KV cache). We clarify that below reported VoiceCraft results are based on a $\textbf{single}$ inference pass. This methodological choice enables a direct and fair comparison of architectural efficiency, isolating the inherent speed and resource advantages of our hybrid Mamba–Transformer design against the Transformer-decoder baseline. The empirical results below provide strong validation of these computational benefits:
>
> 1. $\textbf{Inference Speed:}$ MAVE's linear scaling is confirmed. While slower on very short clips, it becomes faster than VoiceCraft (with KV Cache) for sequences longer than 4 seconds, and the advantage grows substantially with audio length.
>
> 2. $\textbf{GPU Memory:}$ MAVE's memory footprint remains nearly constant (~4.5 GB), regardless of sequence length, thanks to its hybrid architecture. In contrast, VoiceCraft's memory usage grows rapidly, becoming prohibitive for long-form editing.
>
> **Table R4.** Inference Time (seconds)
> | Audio duration (sec) | 2      | 4      | 6      | 8      | 10     | 12     | 14      | 16      | 18      | 20      |
> |:--------------------:|:------:|:------:|:------:|:------:|:------:|:------:|:-------:|:-------:|:-------:|:-------:|
> | VoiceCraft           | **2.0**  | 6.8    | 18.8   | 33.4   | 54.2   | 75.4   | 106.6   | 138.2   | 176.0   | 221.7   |
> | VoiceCraft (KV Cache)| 2.8    | 8.5    | 11.2   | 16.1   | 24.1   | 29.7   | 34.5    | 38.2    | 43.5    | 49.8    |
> | **MAVE** (ours)      | 3.3    | **6.6**  | **9.8**  | **13.1** | **16.1** | **19.6** | **22.6**  | **25.6**  | **29.3**  | **32.5**  |
>
> Crucially, MAVE's practical performance advantage is substantially greater than a direct comparison suggests. To achieve acceptable output quality, VoiceCraft typically requires $N$ sampling runs ($N$=5 in all of our experiments) to mitigate its tendency to generate extended silent segments. This is often accompanied by post-hoc heuristics, such as detecting silence tokens and explicitly down-weighting their probabilities during generation. In contrast, MAVE produces high-quality, silence-free outputs in a single forward pass, making it not only faster but also more robust and deployable in real-world settings. This necessity multiplies its effective inference time by a factor of $N$. We will explicitly mention this critical practical disparity in the revised manuscript.
>
> **Table R5.** GPU Memory Usage (Gb)
> | Audio duration (sec) | 2      | 4      | 6      | 8       | 10      | 12      | 14       | 16       | 18       | 20       |
> |:--------------------:|:------:|:------:|:------:|:-------:|:-------:|:-------:|:--------:|:--------:|:--------:|:--------:|
> | VoiceCraft (KV Cache)| 4.93   | 6.00   | 6.82   | 10.47   | 13.31   | 15.15   | 23.91    | 27.82    | 33.55    | 38.61    |
> | **MAVE** (ours)      | **4.49** | **4.47** | **4.47** | **4.51**  | **4.53**  | **4.56**  | **4.58**   | **4.59**   | **4.65**   | **4.64**   |
>
>
> **References:**
>
> [A] Montreal Forced Aligner: trainable text-speech alignment using Kaldi
>
> [B] UTMOS: UTokyo-SaruLab System for VoiceMOS Challenge 2022

---

### Meta-Review · Area_Chair_xPku · 2025-12-15

**Summary:**

Across the four reviews, MAVE is recognized as a hybrid architecture that combines a Mamba-based state-space model with cross attention for text conditioning. 3 Reviewers generally view the work as innovative, well-presented, and strong in zero-shot TTS performance and efficiency. However, one reviewer is strongly negative, raising concerns about novelty and baseline coverage.  In addition, most reviewers have the concerns on Insufficient baseline comparisons, Efficiency claims not experimentally validated, degradation in long-form TTS and Limitations of the manual-editing interface.

**Reviewer Concerns:**

Insufficient baseline comparisons have been addressed in the rebuttal.

**Reviewer Scores:**

I think the reviewer would have changed their scores.

---

### Decision · Program_Chairs · 2026-01-26

Reject